# Scrub Typhus and Influenza A Co-Infection: A Case Report

**DOI:** 10.3390/pathogens14010064

**Published:** 2025-01-13

**Authors:** Chie Yamamoto, Ayano Maruyama, Jun Munakata, Tasuku Matsuyama, Keitaro Furukawa, Ryosuke Hamashima, Motohiko Ogawa, Yuki Hashimoto, Akiko Fukuda, Tohru Inaba, Yoko Nukui

**Affiliations:** 1Department of Infection Control and Laboratory Medicine, Kyoto Prefectural University of Medicine, Kyoto 602-8566, Japan; k-furu@koto.kpu-m.ac.jp (K.F.); hamashim@koto.kpu-m.ac.jp (R.H.); inaba178@koto.kpu-m.ac.jp (T.I.); y-nukui@koto.kpu-m.ac.jp (Y.N.); 2Department of Dermatology, Graduate School of Medical Science, Kyoto Prefectural University of Medicine, Kyoto 602-8566, Japan; 3Department of Cardiovascular Medicine, Graduate School of Medical Science, Kyoto Prefectural University of Medicine, Kyoto 602-8566, Japan; 4Department of Emergency Medicine, Kyoto Prefectural University of Medicine, Kyoto 602-8566, Japan; 5Department of Virology 1, National Institute of Infectious Diseases, Tokyo 162-8640, Japan

**Keywords:** co-infection, influenza A, scrub typhus

## Abstract

Scrub typhus, caused by *Orientia tsutsugamushi*, is a neglected and reemerging disease that causes considerable morbidity and mortality. It now extends beyond the Tsutsugamushi Triangle, the region wherein it has traditionally been endemic. Influenza has also resurged since the infection control measures against COVID-19 were relaxed. A few cases of scrub typhus and influenza co-infection have been reported. Herein, we report the case of a 74-year-old woman with fever and upper respiratory symptoms diagnosed with influenza A and treated with oseltamivir; however, her fever persisted, and she developed respiratory failure, liver dysfunction, headache, diarrhea, and an erythematous skin rash. She lived in a forested area where scrub typhus was endemic and worked on a farm. Physical examination revealed an eschar on her posterior neck, and she was diagnosed with scrub typhus and influenza A co-infection. After minocycline treatment, her symptoms improved within a few days. This is the first reported case of scrub typhus and influenza A co-infection in Japan. This case illustrates that co-infection should be suspected in patients with fever persisting after their initial infection has been treated and that in patients living in endemic areas, scrub typhus can occur concurrently with influenza. The symptoms of scrub typhus are flu-like and nonspecific, which may delay diagnosis and treatment.

## 1. Introduction

Scrub typhus is a potentially life-threatening zoonotic rickettsiosis caused by the intracellular bacterium *Orientia tsutsugamushi* and transmitted by trombiculid mite larvae [1]. It is endemic in the “Tsutsugamushi Triangle” region bounded by the Russian Far East, Japan, Afghanistan, and Australia. Here, approximately 2 billion people are at risk of infection, approximately 1 million people contract the disease, and 150,000 people die of the disease annually [2,3]. Recently, scrub typhus has been reported outside this region, in Africa and the Middle East, and has been identified as a neglected but reemerging disease [3]. In endemic areas, exposure to high ambient temperature, farming activities, and forests are risk factors for scrub typhus [4]. The proposed pathogenic mechanism is endothelial damage caused by vasculitis, as *O. tsutsugamushi* proliferates in vascular endothelial cells and macrophages, triggering a release of inflammatory cytokines and causing microvasculitis and perivasculitis [5,6]. Fever, skin rash, and an eschar are the three primary clinical manifestations. Other clinical manifestations include headache, fatigue, gastrointestinal symptoms, myalgia, arthralgia, lymphadenopathy, and hepatosplenomegaly [1]. Of these, the finding of an eschar has high specificity (98.9%) and significant diagnostic value. However, eschars occur only in approximately 55% of cases—although this number varies by reports—and the nonspecific nature of the other clinical manifestations often hinders diagnosis [7]. Severe cases can present with complications such as acute respiratory distress syndrome (ARDS), disseminated intravascular coagulation, septic shock, liver dysfunction, thrombocytopenia, pneumonia, acute kidney injury, myocarditis, and meningoencephalitis [1]. Without appropriate treatment, the case fatality rate in severe cases is 30–70%, and in cases with multiple organ failure, the case fatality rate is 24%, even with appropriate treatment [1,8]. Patients with scrub typhus are often co-infected with other infectious diseases, given its endemic area and seasonality. Furthermore, influenza has resurged since the infection control measures against COVID-19 were relaxed [9]. Herein, we present a case of scrub typhus and influenza A co-infection with an aim to illustrate the difficulties with diagnosis and treatment of scrub typhus and influenza co-infections.

## 2. Case Report

A 74-year-old woman presented with the primary complaint of fever, sore throat, and cough. She had a history of chronic heart failure, hypertension, and dyslipidemia. She did not drink alcohol or smoke, lived in the forest area of the southeastern Shiga Prefecture in Japan, and worked on the farm adjacent to her house.

In early November 2023 (day 0), she presented with a fever of 39 °C, a sore throat, and dry cough. On day 2, she was diagnosed with influenza A by her local physician, based on a rapid influenza antigen test (Rapid Testa FLU & SARS-CoV-2, SEKISUI MEDICAL Co., Ltd., Tokyo, Japan) using a nasopharyngeal swab, and started on antipyretics and oseltamivir for 5 days. However, the fever persisted; she developed an erythematous rash distributed mainly on her trunk and developed shortness of breath on day 6. Her physician admitted her to hospital on day 7. She was treated with ceftriaxone, but the fever did not resolve, and she developed liver dysfunction, headache, and diarrhea.

Consequently, she was transferred to our hospital on day 9. On admission, her Glasgow Coma Scale score was E4V4M6, with a body temperature of 38.9 °C, heart rate of 120 beats/min, blood pressure of 120/66 mmHg, SpO_2_ of 89% (breathing ambient air), and a respiratory rate of 20 breaths/min. Auscultation revealed coarse crackles bilaterally in the lower lung fields, and bilateral pitting edema was present in the lower legs. She had a generalized erythematous rash without infiltration on the face, trunk, and limbs.

Her blood test results on admission to our hospital are shown in Table 1.

Sputum examination revealed no significant bacteria, and two sets of four blood cultures tested negative. An electrocardiogram showed rapid atrial fibrillation, and a chest radiograph showed extensive infiltrative shadows bilaterally, mainly in the lower lung fields.

The patient’s clinical course is summarized in Figure 1. After transfer, the patient was treated with supportive care for respiratory failure and exacerbation of her chronic heart failure, while we investigated the cause of her prolonged fever and skin rash.

Physical examination was repeated on day 10, and an eschar with an ulcer, 5 × 6 mm in size, with a black crust and surrounding erythema was observed on her left posterior neck region (Figure 2). She was uncertain when the mite bite occurred.

Duplex real-time polymerase chain reaction of an eschar sample (Duplex TaqMan PCR for *O. tsutsugamushi* and *R. japonica*; Thermo Fisher Scientific, Inc., Waltham, MA, USA) tested positive for *O. tsutsugamushi.* Furthermore, paired serum samples collected on days 11 and 25 showed a significant increase in serum antibody titers against *O. tsutsugamushi* on immunofluorescence (Appendix A). Based on these findings, the patient was diagnosed with scrub typhus and influenza A co-infection.

Phylogenetic tree analysis of the gene sequence of the 56 kDa type-specific antigen obtained from the eschar revealed that the sequence was the closest to the Kawasaki type (Appendix A). Nucleotide sequence data reported are available in the DDBJ databases under the accession number LC848967. Accordingly, the patient was treated with minocycline, 100 mg twice a day, starting on day 10; the fever resolved on day 11. Subsequently, the skin rash, liver dysfunction, respiratory and heart failure resolved, and lung shadows on chest radiography improved (chest radiographs taken on day 9 and 18 are shown in Figure 1). The patient continued to improve. Minocycline was discontinued after 10 days, and the patient was discharged on day 23.

## 3. Discussion

Co-infections of scrub typhus with malaria, dengue fever, chikungunya, typhoid fever, leptospirosis, mycoplasma, Q fever, Japanese encephalitis, Japanese spotted fever, severe febrile thrombocytopenia syndrome, hepatitis E, *Nocardia*, COVID-19, and influenza have been reported [10,11,12,13,14,15,16,17,18]. As the symptoms of scrub typhus—other than the eschar—and these co-infections are similar and nonspecific, the diagnosis of scrub typhus is often delayed [15,16,17,18]. The effect of co-infections varies. In cases of co-infection with malaria or dengue fever, the disease tends to be milder than in cases of scrub typhus alone [11]. By contrast, co-infection with chikungunya tends to lead to more severe disease, with a higher frequency of headache, abdominal pain, dyspnea, convulsions, and splenomegaly [12]. The reason for the difference in the clinical course of different co-infections remains unclear.

At the initial visit, our patient was diagnosed with and treated for influenza A, which did not improve. She subsequently developed respiratory failure, a skin rash, liver dysfunction, headache, and diarrhea. The discovery of an eschar led to the diagnosis of scrub typhus co-infection.

According to a study by Jhuria et al. [17], 4.7% of patients in scrub typhus-endemic areas who require hospitalization for influenza are co-infected with scrub typhus. Approximately 400–500 cases of scrub typhus are reported per year in Japan, and November, when the present case occurred, is also the active season for the larvae of *Leptotrombidium scutellare*, a species of mites that transmit the Kawasaki and Irie types of *O. tsutsugamushi* [19]. In November 2023, influenza was also prevalent in Japan, resulting in an overlap of the scrub typhus and influenza epidemic periods [20]. Such co-infection of scrub typhus and influenza is not rare in endemic areas. However, detailed information about this co-infection is not available, warranting further studies. Table 2 summarizes the five cases of scrub typhus and influenza co-infection reported to date, including this case [17,18].

All patients had fever and cough, with durations lasting from 5 to 10 days, with that in our case being the longest. By contrast, only one and three patients presented with a skin rash and an eschar, respectively. Fever, skin rash, and an eschar are the three main signs of scrub typhus. The blood test results revealed liver dysfunction in all patients. Furthermore, except for Case 4, all patients had Sequential Organ Failure Assessment (SOFA) scores of 2 or higher and sepsis, and three patients had respiratory failure, one of whom required invasive positive-pressure ventilation. Bilateral infiltrating shadows in the lung fields were observed on chest radiography in all three patients with respiratory failure. The main cause of respiratory failure in our patient was believed to be an exacerbation of chronic heart failure or rapid atrial fibrillation triggered by the infection; she was not diagnosed with ARDS. Regarding the timing of co-infection diagnosis, two of the five patients were diagnosed with scrub typhus at the same time as influenza, whereas three—including our patient—were diagnosed with influenza first and scrub typhus subsequently. In our patient, her prolonged fever and skin rash were initially suspected to be caused by influenza or an allergy to drugs, such as oseltamivir and antipyretics. However, we repeated the patient’s physical examination because her residence was in a forested area, where scrub typhus had occurred previously, and she worked on a farm. We found the eschar on day 10, which led to the scrub typhus diagnosis, introduction of appropriate treatment, and her subsequent recovery. Unlike this case and another case in which an eschar was recognized by repeating the physical examination, the diagnosis in another case was made by measuring serum antibody titers. All five patients, including the three who were diagnosed late, recovered after treatment with tetracycline antibiotics and azithromycin.

This is the first reported case of scrub typhus and influenza A co-infection in Japan. This case demonstrates that patients with fever may have more than one type of infection concurrently and that, in an endemic area, scrub typhus can be present with other infectious diseases, including influenza. Scrub typhus diagnosis tends to be delayed because, other than the eschar, the symptoms are nonspecific. Delays in diagnosis lead to delays in commencing the appropriate antibiotic treatment, which may in turn increase the disease severity. Understanding the prevalence of infectious diseases in the patient’s community and re-confirming the patient’s history and performing a detailed physical examination are important when clinical manifestations such as fever do not improve.

## Figures and Tables

**Figure 1 pathogens-14-00064-f001:**
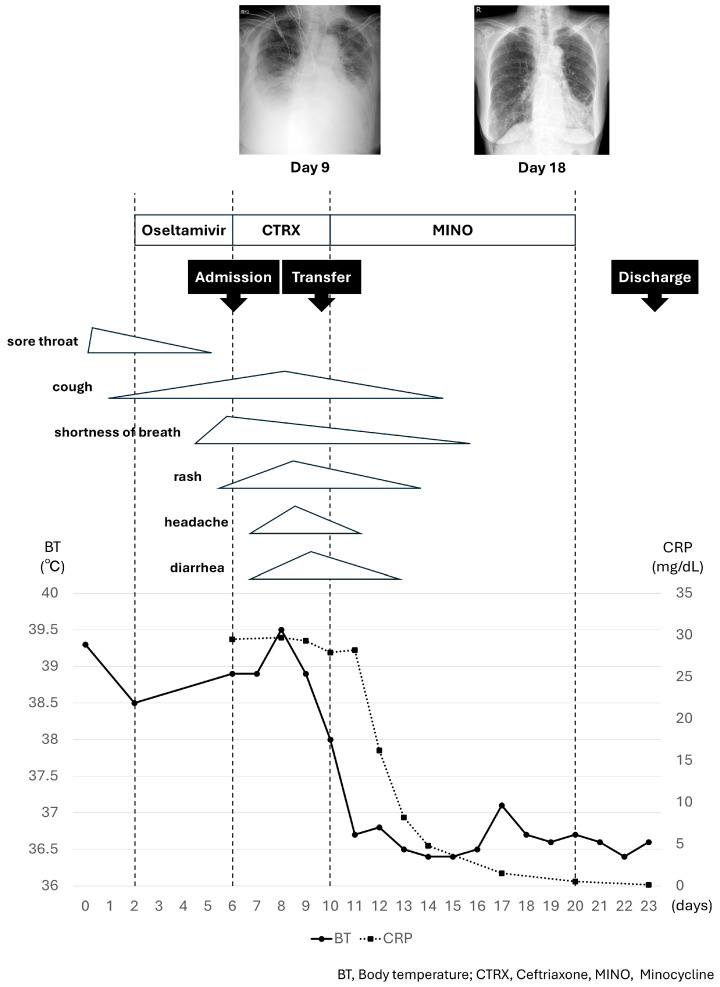
The patient’s clinical course.

**Figure 2 pathogens-14-00064-f002:**
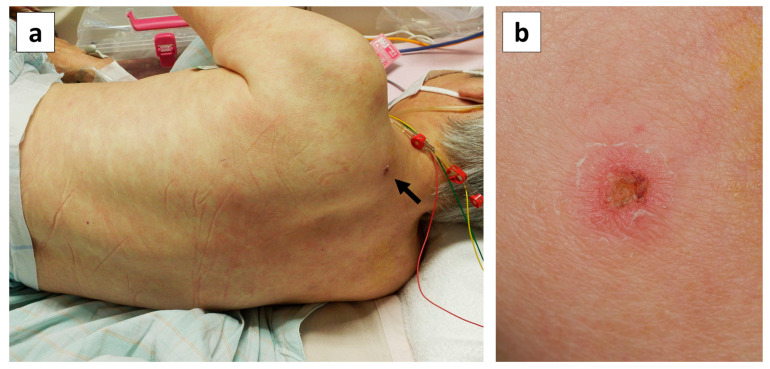
Skin findings on the patient’s back on day 10 of illness. (**a**) Erythema maculatum without infiltration showing a partial tendency to fuse on the trunk and limbs, and an eschar on the left posterior region of the neck (black arrow). (**b**) Close-up photograph of the eschar on the left posterior neck region, showing an ulcer with a black crust, 5 × 6 mm in size, with surrounding erythema.

**Table 1 pathogens-14-00064-t001:** The patient’s blood test results on admission to our hospital (day 9).

Parameter (Units)	Value	Reference Range
WBC (cells/µL)	18,300	4500–11,000
Neutrophils (%)	86.4	40–80
Lymphocytes (%)	11.3	20–40
Monocytes (%)	2.0	2–8
Eosinophils (%)	0.1	1–4
Basophils (%)	0.2	<1
RBC (×10^4^ cells/µL)	399	380–520
Hemoglobin (g/dL)	10.8	12–15
Hematocrit (%)	42.0	36–48
MCV (fL)	81.5	80–100
MCH (pg)	27.1	27–33
MCHC (%)	33.2	32–36
Platelets (×10^4^ /µL)	24.4	15.7–37.1
Total protein (g/dL)	5.6	6–8
Albumin (g/dL)	2	3.4–5.4
Total bilirubin (mg/dL)	0.2	0.2–1.3
AST (U/L)	64	10–36
ALT (U/L)	41	5–38
LDH (U/L)	234	135–214
ALP (U/L)	105	44–147
γ-GTP (U/L)	45	8–38
BUN (mg/dL)	27	8–23
Creatinine (mg/dL)	1.71	0.6–1.1
Na (mEq/L)	130	135–145
K (mEq/L)	3.3	3.6–5.2
Cl (mEq/L)	96	96–106
CRP (mg/dL)	29.3	<0.3
Glucose * (mg/dL)	111	70–99
HbA1c (%)	6.0	<5.7
PT (s)	15.8	11–13.5
PT-INR	1.24	0.8–1.1
APTT (s)	43.8	25–35
FDP (µg/mL)	6.2	<10
BNP (pg/mL)	433.2	<100

RBC, red blood cell; MCV, mean corpuscular volume; MCH, mean corpuscular hemoglobin; MCHC, mean corpuscular hemoglobin concentration; AST, aspartate aminotransferase; ALT, alanine transaminase; γ-GTP, gamma-glutamyl transpeptidase; ALP, alkaline phosphatase; PT-INR, prothrombin time-international normalized ratio; APTT, activated partial thromboplastin time; BNP, B-type natriuretic peptide; BUN, blood urea nitrogen; CRP, C-reactive protein; FDP, fibrin/fibrinogen degradation products; HbA1c, glycated hemoglobin; LDH, lactate dehydrogenase; PT, prothrombin time. * Random glucose testing.

**Table 2 pathogens-14-00064-t002:** Clinical features, laboratory findings, and management of patients with co-infection of scrub typhus and influenza, as reported in the literature.

No.	Author (Year) [Ref.]	Country	Type of Influenza	Age (Years) Sex	Underlying Conditions	Clinical Features						Investigations				Organ Dysfunction				Therapy	Outcome
						Fever Duration (Days)	Rash	Eschar	Headache	Cough	SOB	Plt × 10^4^ /µL	AST U/L	ALT U/L	CXR	PaO_2_/ FiO_2_	Hypo- Tension	GCS	SOFA Score		
1	Jhuria et al. 2020 [17]	India	H1N1	30 F	NA	7	−	+	+	+	+	13.5	104	56	Bilateral infiltrates	110	No	15	8	NPPV OTV (5 d) CTRX + AZM (7 d)	Recovered
2	Jhuria et al. 2020 [17]	India	H1N1	23 M	NA	5	−	−	+	+	+	11.8	148	41	Bilateral infiltrates	200	No	15	10	IPPV OTV (5 d) CTRX + DOXY (7 d)	Recovered
3	Jhuria et al. 2020 [17]	India	H1N1	28 F	Pregnancy	5	−	−	−	+	−	15.4	123	134	Normal	414	No	15	4	OTV (5 d) CTRX + AZM (7 d)	Recovered
4	Ahn et al. 2011 [18]	South Korea	H1N1	53 F	Bronchi- ectasis	5	−	+	−	+	−	13.1	635	788	Normal	NA	No	15	NA	OTV zanamivir DOXY	Recovered
5	This case	Japan	Type A	74 F	CHF, HT Dyslipidemia	10	+	+	+	+	+	24.4	64	41	Bilateral infiltrates	281	No	14	4	OTV CTRX (5 d) MINO (10 d)	Recovered

+, present; −, absent; ALT, alanine aminotransferase; AST, aspartate aminotransferase; AZM, azithromycin; CHF, congestive heart failure; CTRX, ceftriaxone; CXR, chest X-ray; DOXY, doxycycline; GCS, Glasgow Coma Scale; HT, hypertension; IPPV, invasive positive pressure ventilation; MINO, minocycline; NA, not applicable or not available; NPPV, noninvasive positive pressure ventilation; OTV, oseltamivir; Plt, platelet count; SOB, shortness of breath; SOFA, Sequential Organ Failure Assessment.

## Data Availability

The original contributions presented in this study are included in the article/Appendix A. Further inquiries can be directed to the corresponding authors.

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
