# Peer review of "Scrub Typhus and Influenza A Co-Infection: A Case Report"

_pathogens, 2025, doi:10.3390/pathogens14010064_

Round 1
Reviewer 1 Report
Comments and Suggestions for Authors
General comments
Would the authors recommend that a tetracycline also be administered along with oseltamivir early in cases of similar patient presentation in endemic areas for scrub typus, with or without eschar, when a rapid influenza test is positive?
Specific comments
Lines 55-56 - The authors refer to a resurgence of influenza due to a relaxation of COVID-19 measures. Was the rate of influenza vaccination in the Shiga Prefecture before and after COVID-19 different? If similar, what other factors could account for an increased incidence of influenza?
Figure 1 - Recommended stating the day each X-ray was taken if they are aligned with the course of treatment.
Line 95 - Is it suspected that the mite was acquired while walking through brush vegetation on the way to cultivated fields where the patient worked?
Line 146 - The authors note that “Sequential Organ Failure Assessment 146 (SOFA) scores of 2 or higher and sepsis were present in all but case #4. Was this attributed to this patient being placed on doxycycline w/o prior treatment with ceftriaxone?
Table 2 - It would be additive to indicate the serologic subtype designation of influenza A (e.g., H1N1, H3N1). This would be specified in the package insert of the rapid influenza antigen test performed on day 2.
Reviewer 2 Report
Comments and Suggestions for Authors
1) Is this the first report of co-infection with Scrub typhus and Influenza in Japan? If it is not, then the author can add those case reports (if previously reported in Japan) in Table 2.
2) The glucose data presented in Table 1 lacks clarity, as it does not specify whether it refers to fasting or postprandial or random glucose test report?
3) On day 9, when the patient was transferred to another hospital (author’s hospital), the recorded fever was 38.9 C; however, the figure 1 presented indicates a temperature of 38.5 C. It is advisable to amend this figure for accuracy.
4) On examination, the author noted the presence of bilateral pitting edema in the lower legs of the patient. What is the significance of this observation? Moreover, might it have been exacerbated by Scrub typhus or influenza?
5) During the evaluation on day 9, the author noted a generalized erythematous rash without infiltration on the face, trunk and limbs, but did not observe any eschar. However, upon re-evaluation on day 10, the author noted the presence of eschar. This raises the question of why eschar was not detected on day 9.
Comments on the Quality of English LanguageNone
